# Coupling sensor to enzyme in the voltage sensing phosphatase

Yawei Yu [1], Lin Zhang[1], Baobin Li [1,5], Zhu Fu [1], Stephen G. Brohawn [1,2,3] & Ehud Y. Isacoff [1,2,3,4] ✉

Voltage-sensing phosphatases (VSPs) dephosphorylate phosphoinositide (PIP) signaling lipids in response to membrane depolarization. VSPs possess an S4-containing voltage sensor domain (VSD), resembling that of voltage-gated cation channels, and a lipid phosphatase domain (PD). The mechanism by which voltage turns on enzyme activity is unclear. Structural analysis and modeling suggest several sites of VSD-PD interaction that could couple voltage sensing to catalysis. Voltage clamp fluorometry reveals voltage-driven rearrangements in three sites implicated earlier in enzyme activation—the VSD-PD linker, gating loop and R loop—as well as the N-terminal domain, which has not yet been explored. N-terminus mutations perturb both rearrangements in the other segments and enzyme activity. Our results provide a model for a dynamic assembly by which S4 controls the catalytic site.

Voltage-sensing phosphatase (VSP) is activated upon membrane depolarization and dephosphorylates various phosphoinositide (PIP) signaling lipids, connecting the electrical activity of a cell to its intracellular signaling[1–5]. VSP contains a transmembrane voltage sensor domain (VSD) that controls a cytosolic lipid phosphatase domain (PD), which has homology to phosphatase and tensin homolog (PTEN)[1,6–11]. The VSD is similar in structure and voltage sensing mechanism to that of voltage-gated cation channels, with an arginine-rich and hence positively charged S4 transmembrane segment that moves outward in response to membrane depolarization[12–14]. This rearrangement turns on enzyme activity in VSPs, while in voltage-gated cation channels it opens the pore gate.

X-ray crystallography on the isolated cytoplasmic domain of *Ciona intestinalis* VSP (Ci-VSP) has captured it in three major poses, which primarily differ in the conformation of two segments: the "gating loop", which either obstructs the catalytic site or swings away[15], and another loop, which is flanked by two arginines (the "R-loop") and includes hydrophobic residues that have been proposed to form weak interaction with the membrane[16]. Two models have been proposed for how the voltage sensor is coupled to catalytic activity. The "gated-access" model[9,15] proposes that the gating loop obstructs access to the

catalytic site at negative voltages and swings away at positive voltages to open access to substrate, thereby turning on enzyme activity. In this model, the movement of S4 is transmitted to the PD via the interaction between the gating loop and the intracellular VSD-PD linker (the "linker"), which immediately follows S4. On the other hand, the "PD displacement" model[16,17] proposes that the S4-linker interacts with the R-loop and that outward motion of S4 pulls the R-loop "up" close to the membrane, bringing the catalytic site within reach of PIP headgroups, and thus turning on enzymatic activity.

However, there is thus far limited evidence for voltage-dependent movement in these key intracellular parts of enzymatically active VSPs and it remains unclear how voltage change induces conformational rearrangement of the catalytic site. To elucidate this, we applied voltage clamp fluorometry (VCF)[18] in Ci-VSP to study conformational motions in different parts of the protein, by either attaching tetramethylrhodamine maleimide (TMRM) to an introduced cysteine in the extracellular S3-S4 loop[12] or labeling intracellular segments with the fluorescent unnatural amino acid 3-(6-acetylnaphthalen-2-ylamino)−2-aminopropionic acid (ANAP)[19–21]. We observed voltage-driven rearrangements in the linker, R-loop, gating loop and catalytic site. All of these rearrangements, except in the S4-proximal segment of the

[1]Department of Molecular and Cell Biology, University of California, Berkeley, California, USA. [2]Helen Wills Neuroscience Institute, University of California, Berkeley, California, USA. [3]The California Institute for Quantitative Biosciences (QB3), University of California, Berkeley, California, USA. [4]MBIB Division, Lawrence Berkeley National Laboratory, Berkeley, California, USA. [5]Present address: Zhongshan Hospital, Institute for Translational Brain Research, State Key Laboratory of Medical Neurobiology, MOE Frontiers Center for Brain Science, Fudan University, Shanghai, China. ✉e-mail: ehud@berkeley.edu

linker, were impacted by mutations at residues that were previously suggested to have a role in voltage-activity coupling[9,15]. However, neither protein motion nor enzymatic activity was completely disrupted by these mutations, suggesting that other determinants may also contribute to VSD-PD coupling.

To identify additional coupling determinants, we performed cryogenic electron microscopy (cryo-EM) on the *Danio rerio* VSP (Dr-VSP) and obtained two structures: a low-resolution monomer structure in a nanodisc and a high-resolution PD dimer structure. An AlphaFold model of Dr-VSP, which is predicted to be structurally very similar to Ci-VSP, could be docked into the low-resolution nanodisc structure. The model predicted that the membrane-proximal (near S1) portion of the N-terminal domain is structured and that it interacts with both the linker and the gating loop. We test this experimentally and find that the membrane-proximal N-terminus undergoes a voltage-dependent rearrangement, which is perturbed by mutations in the linker, the R-loop and the gating loop. Complementary to this, mutations of the membrane-proximal portion of the N-terminus alter voltage-dependent motions in S4, the distal linker, the gating loop and the catalytic site, and also perturb enzyme activity. Our observations suggest a three-part assembly for transmitting the voltage-driven motion of S4 to the catalytic site: the membrane-proximal N-terminus of the VSD, the distal (near-PD) end of the inter-domain linker and the PD gating loop.

## Results

### Voltage-dependent rearrangements in the linker, R-loop and gating loop

Our earlier X-ray crystallography study[15] revealed several conformations of the isolated cytoplasmic domain that showed large differences in the pose of two segments (Supplementary Movie-1): the "gating loop" (Ci-VSP residues 398-413), which alternated between conformations that obstruct and allow access to the catalytic site, and the "R loop" (Ci-VSP residues 281-286), which has been proposed to loosely associate with the membrane in a "hinge-like" motion[16]. In the full-length functioning protein, voltage-driven conformational change has been detected at residue 401[20], located at the beginning of the gating loop and next to D400, which was proposed to form an electrostatic interaction with the VSD-PD linker[15]. However, protein motion has not been examined in the linker, near the glutamate gate (E411) of the gating loop, in the R-loop or at the catalytic site. To understand how voltage-driven S4 movements are transmitted to the catalytic site, we used tetramethylrhodamine maleimide (TMRM) attached to the extracellular end of S4 as a reporter for its movements. We also introduced the fluorescent unnatural amino acid (fUAA) L-3-(6-acetylnaphthalen-2-ylamino)−2-aminopropanoic acid (ANAP)[21] individually in the key intracellular segments. Since the emission maximum of the ANAP fluorescence spectra shifts to longer wavelengths in more polar solvent[21,22], we monitored ANAP fluorescence simultaneously over two wavelength ranges 420−460 nm and 460−500 nm, ratiometric measurement of which could provide information on the nanoenvironment changes around the labeling site due to local protein motion[20].

As shown earlier[12], depolarizing voltage steps induced a change of fluorescence (ΔF) from TMRM attached covalently to an introduced cysteine at residue 214 located at the external end of S4 (G214C) (Fig. 1A). Voltage-dependent ΔFs of ANAP were also observed when it was incorporated at the beginning of the linker (residue 243 in the helical segment near S4), at the opposite end of the linker (residue 261 in the beta hairpin near the PD), next to the R-loop (residue 290), in the gating loop near the E411 glutamate gate (residue 409), as well as at a C2 domain site that points into the catalytic pocket (residue 522) (Fig. 1B−D, F, G). ANAP ΔFs observed from residue 401, near the beginning of the gating loop (Fig. 1E), and a C2 domain residue 555 (Fig. 1H), the homolog of which was suggested to associate with the

membrane in PTEN[23], were similar as observed previously[20]. These results demonstrate that, in the full-length protein, membrane depolarization drives structural changes in the linker, R loop and gating loop. The voltage dependence of these rearrangements suggests that they have a role in PD activation. Compared to labeling sites 214 in S4 and 243 at the beginning of the linker, the voltage dependence of dye fluorescence (F-V) was shifted positively in the distal linker, gating loop and catalytic site (Fig. S1), suggesting that outward movement of S4 drives subsequent rearrangements that turn on enzyme activity.

We tested the catalytic function of Ci-VSPs with ANAP incorporated at different sites using the PI(4,5)P$_2$-activated inward rectifier potassium channel, Kir2.1 R228Q (Kir2.1Q), as a readout of PI(4,5)P$_2$→PI(4)P 5' phosphatase activity[1,12] (Supplementary Fig. S2A; see "Methods"). The catalytic activity of the ANAP-incorporated Ci-VSPs was ≥75% of wildtype at most of the sites but was substantially compromised at two sites: at linker residue 261 and, not surprisingly, at gating loop residue 409, which is located close to the E411 gate (Supplementary Fig. S2B, C). There was little perturbation at other sites in the same segments: linker residue 243 and gating loop residue 401.

### Linker-gating loop interaction in coupling S4 motion to PD rearrangements and catalysis

To test the model that the linker couples S4 motion to the rearrangements in the gating loop, and also to assess its role in coupling to movement in the R-loop, we examined the impact of mutating lysine 252 in the linker and aspartate 400 in the gating loop. These residues have been shown to form a salt bridge in the crystal structure of the isolated cytoplasmic domain and mutations of these residues affect the voltage-dependent enzyme activity[9,15]. TMRM labeling of ANAP-incorporated Ci-VSP showed that the K252E, K252Q and D400R mutations did not alter expression at any of the ANAP labeling sites (Supplementary Fig. S3), permitting us to assess the effects of these mutations on both ΔF amplitude and its direction. K252E had little impact on the ΔF in the proximal linker (Fig. 1B) but had a major impact in the distal linker, R-loop, gating loop, catalytic site and C2 domain (Fig. 1C–H). K252Q and D400R had similar effects to K252E, but the effects from D400R were milder (Fig. 1 and Supplementary Fig. S4). At some sites, the mutants flipped the direction of the ΔF (Fig. 1I). To gain insight into how mutations affect the structure at each ANAP labeling site in the resting state, we compared the fluorescence ratio from the two emission channels (R = F$_{460-500}$ / F$_{460-500}$ + F$_{420-460}$) at the holding voltage −100 mV (R$_{baseline}$). We did not observe any effect of the mutations on R$_{baseline}$, but low signal to noise ratio due to fluorescence background of the oocytes and/or free ANAP may have obscured a difference. We therefore do not draw a conclusion about this point. However, the analysis of the change of fluorescence ratio between R$_{baseline}$ (−100 mV) and R$_{test}$ (+200 mV) normalized to R$_{baseline}$ (R$_{test}$ − R$_{baseline}$ / R$_{baseline}$, or ΔR / R) eliminates the influence of the background and thus describes the structural changes that each ANAP site undergoes upon activation. The amplitude of ΔR / R indicates the extent of the spectral shift and the sign of the ΔR / R shows the direction of the shift. An increase of ΔR / R, such as what we observed with ANAP introduced at position 401 (gating loop residue next to linker-gating loop coupling residue D400) in WT Ci-VSP, indicates a spectral shift to longer wavelengths and exposure to more polar solvent[21,22]. This is consistent with the overlay of previous X-ray crystal structures of the isolated Ci-VSP PD (Supplementary Movie-1), which shows that the side chain of F401 moves to a more solvent exposed environment in the transition from the "closed" state (3V0F, the catalytic site "closed" / E411 blocked conformation) to the "activated" state (3V0H, the catalytic site bound confirmation)[15]. The amplitude of the ΔR / R (i.e., extent of the voltage-dependent spectral shift) was strongly affected by the D400R mutation in the gating loop and the catalytic sites and much less so elsewhere in the protein. Mutations of K252 had even larger effects on the spectral shift (Fig. 1J). The

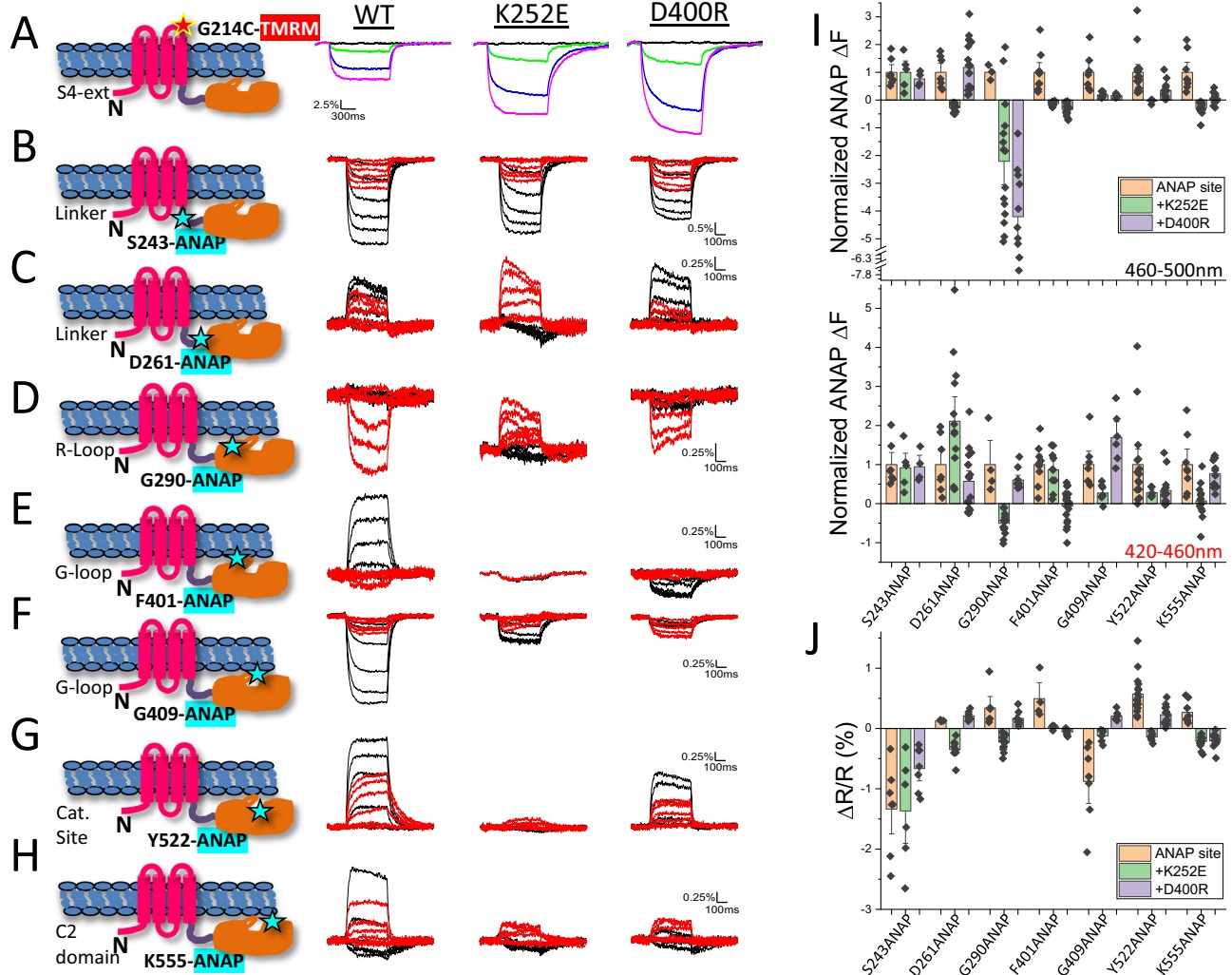

**Fig. 1 | The linker, R loop, gating loop and catalytic site undergo voltage-driven conformational changes and mutation of K252E and D400R strongly affect these movements. A** Representative fluorescence traces from TMRM-labeled G214C of WT Ci-VSP, with K252E, and with D400R at 200 mV (magenta), 80 mV (blue), 20 mV (green) and −100 mV (black). **B–H** Superposition of representative fluorescence traces from ANAP incorporated at proximal linker 243 (**B**), distal linker 261 (**C**), near R-loop 290 (**D**), gating loop 401 (**E**) and 409 (**F**), catalytic site 522 (**G**) and C2 domain 555 (**H**). Fluorescence was collected simultaneously from 420–460 nm (red) and 460–500 nm (black) using protocol 2 (See "Methods"). Voltage steps are 200 mV, 160 mV, 120 mV, 80 mV, 40 mV and 0 mV. Only traces at 200 mV are shown for F401ANAP-K252E due to its small amplitude. **I** Comparison of fluorescence amplitude of ANAP at different sites to that with K252E and D400R

from 460–500 nm (top) and 420–460 nm (bottom) (mean ± s.e.m.; $n$ = 7, 5, 4, 7, 13, 17, 3, 15, 10, 9, 10, 19, 7, 8, 6, 16, 6, 14, 8, 14, 12 from left to right respectively). **J** Comparison of fluorescence ratio change between the 460–500 nm and the 420–460 nm emission channels at different sites to that with K252E or D400R (mean ± s.e.m.; $n$ = 7, 6, 7, 4, 9, 7, 6, 15, 9, 4, 7, 7, 7, 6, 6, 16, 7, 14, 8, 14, 13 from left to right respectively). The cartoon in each panel indicates the individual labeling site. Recordings on individual sites with and without mutations were done on the same batch of oocytes and on the same day, and the same batch of injected oocytes were labeled with TMRM under the same conditions. Each n represents an independent measurement from an individual oocyte. The scales of X and Y axis are the same for each ANAP labeling site and that with K252E and D400R.

reductions in ΔF amplitude and the shifts of ANAP spectrum suggest that the mutations changed either the local environment of the labeling sites or the nature of the rearrangement.

To further probe the role of the K252-D400 (linker to gating loop) connection, we measured the impact of these mutations on S4 movement, which reflects "reverse coupling" from the PD to S4. In WT Ci-VSP, TMRM attached to a cysteine introduced at residue 208 (Q208C-TMRM) in the extracellular S3-S4 loop has a 3-component ΔF: an F1 phase, consisting of a small increase in fluorescence with depolarizations from −100 mV up to 0 mV, followed by an F2 phase, consisting of a large decrease in fluorescence between 0 and 100 mV, and then an F3 phase, consisting of an increase in fluorescence at above 100 mV[9,12] (Fig. 2A, B). Deletion of the PD after linker residue 240 (ΔPD) eliminated F3 (Fig. 2A, B). F3 was also eliminated by K252E

(Fig. 2A, B), consistent with a severe disruption of VSD-PD coupling and its dramatic effects on rearrangements in the gating loop and catalytic site. D400R also had milder effects on S4 motion than K252E (Fig. 2A, B).

We next tested the impact of K252E and D400R on "forward coupling" in a functional assay of voltage-activated catalysis, using our FRET reporters F-PLC and F-TAPP whose FRET increases when binding to PI(4,5)P$_2$ and PI(3,4)P$_2$, respectively[24]. F-PLC and F-TAPP allow monitoring of the accumulation of these PIP$_2$s, as Ci-VSP removes either the 3' or 5' phosphate from PI(3,4,5)P$_3$, and their depletion, as Ci-VSP removes either the 5' phosphate from PI(4,5)P$_2$ or the 3' phosphate from PI(3,4)P$_2$, respectively (Fig. 2C)[24]. The function of both the K252E and D400R mutants was severely compromised and the PIP$_2$→PIP activity−removal of 5' phosphate from PI(4,5)P$_2$ (the late decrease

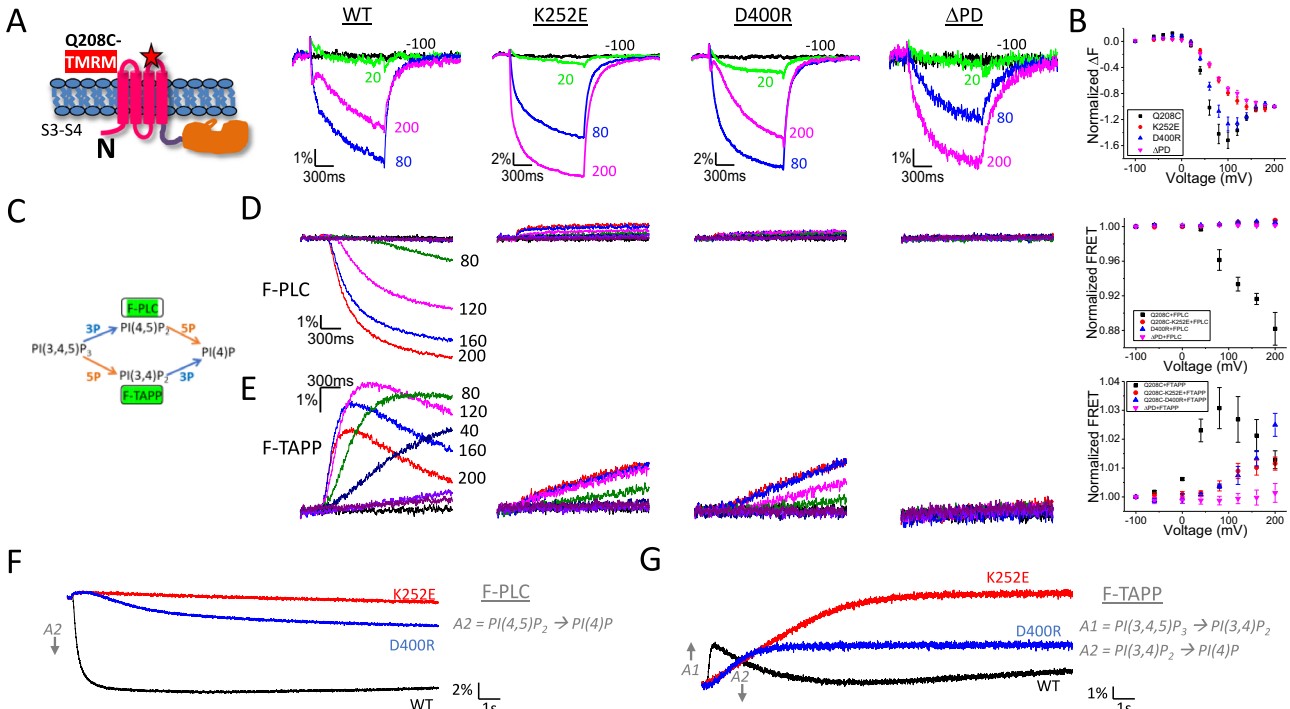

**Fig. 2 | Mutation of K252E and D400R strongly affect voltage dependent S4 conformational change and enzymatic activity. A** Representative fluorescence traces of TMRM-labeled Q208C of WT Ci-VSP, with K252E, with D400R and the ΔPD mutant (truncated after residue 240) at 200 mV (magenta), 80 mV (blue), 20 mV (green) and −100 mV (black). Cartoon indicates labeling scheme. **B** Comparison of F-V curves obtained from TMRM-labeled Q208C of WT Ci-VSP, with K252E, with D400R and the ΔPD mutant (mean ± s.e.m.; n = 23, 9, 10, 9, respectively). Fluorescence measured at the end of each test voltage was normalized to that obtained at 200 mV. **C** Schematics of the phosphoinositide pathways and substrates sensed by different FRET probes. **D, E** Representative FRET traces from F-PLC (**D**) and F-TAPP (**E**) co-expressed with WT Ci-VSP, K252E, D400R or ΔPD (under the corresponding labels and TMRM F-V curves in (**A**) using the 2-s test protocol (See "Methods") and the comparison of normalized FRET from each condition (mean ± s.e.m.; n = 3, 3, 4, 9 respectively for F-PLC and n = 3, 4, 3, 9 respectively for F-TAPP). Each n represents an independent measurement from an individual oocyte. Some test voltages are shown next to corresponding traces. **F, G** Representative FRET traces from F-PLC (**F**) and F-TAPP (**G**) co-expressed with WT Ci-VSP, K252E and D400R using the long (20 s) test protocol.

phase of F-PLC ΔFRET) and 3' phosphate from PI(3,4)P₂ (the late decrease phase of F-TAPP ΔFRET)—was rendered undetectable in a 2-s depolarizing step (Fig. 2D, E). A long (20 s) depolarizing step protocol showed that the decrease phase of F-PLC ΔFRET was not completely eliminated but was strongly attenuated (Fig. 2F). The two mutants retained severely diminished PIP₃→PIP₂ activity, whereby removal of 5' phosphate from PI(3, 4, 5)P₃ (the early increase phase of F-TAPP ΔFRET) was slowed by ~14-fold by D400R and ~44-fold by K252E (Fig. 2G). We were not able to assess changes in the removal of 3' phosphate from PI(3, 4, 5)P₃ (the early increase phase of F-PLC ΔFRET) because of its small amplitude in WT Ci-VSP. We also confirmed that both mutants expressed similarly compared to the WT, suggesting that the functional impacts of the mutants were not due to altered expression (Fig. S5). Overall, both mutants severely reduced voltage-activated enzymatic activity, but again the impact of K252E was stronger than that of D400R, suggesting that K252 may have an additional interaction partner. This is consistent with the severe reduction of coupling to the protein motions that turn on the enzymatic activity in D400R, and the even stronger impact in mutations of K252.

These observations provide evidence for voltage-dependent rearrangements in the linker, the R-loop and the gating loop, and show that these are associated with protein motions at the catalytic site. The results also suggest that the interaction between linker residue K252 and gating loop residue D400 plays an important role in connecting these protein motions. However, even when K252 or D400 is mutated, some voltage-dependent structural rearrangements and enzyme activity persist, and mutation of K252 has even stronger

impacts than that of D400. These suggest that there are additional connections between S4 and the PD that contribute to coupling the conformational changes of the VSD to those in the PD and to the enzymatic activity.

## Cryo-EM structure and AlphaFold model

In search of additional connection(s) between the VSD and PD, we turned to AlphaFold structural modeling. The AlphaFold model of Ci-VSP showed S4 in the "up-plus" configuration[25] with the gating loop in the "open" conformation (E411 swung out of the way to provide access to the C363 catalytic site) (Fig. 3A), suggesting the activated/enzyme ON state. The S4-proximal half of the linker was helical and situated in close proximity to the R-loop, consistent with the interaction proposed earlier[17]. The distal linker was situated close to the gating loop, consistent with previous crystal structures[15]. The AlphaFold model also predicted a new component that interacts with the key parts of the PD: the N-terminal domain. A short N-terminal segment preceding S1, including a pre-S1 helix, is predicted to be ordered. The beginning of this ordered section is situated in close proximity to the distal linker and the gating loop, suggesting a potential VSD-PD coupling interaction (Fig. 3A; Supplementary Movie-2).

We noted that the confidence score of the AlphaFold model was low in regions that connect the VSD to the PD (Fig. 3B). This motivated us to pursue structural analysis on the VSD-PD arrangement using cryo-EM. Compared to Ci-VSP, *Danio rerio* VSP (Dr-VSP) with the active site cysteine mutated to serine (C302S) showed better protein expression and stability. Dr-VSP also has high sequence identity (48%) with Ci-VSP (Supplementary Fig. S6) and an AlphaFold model

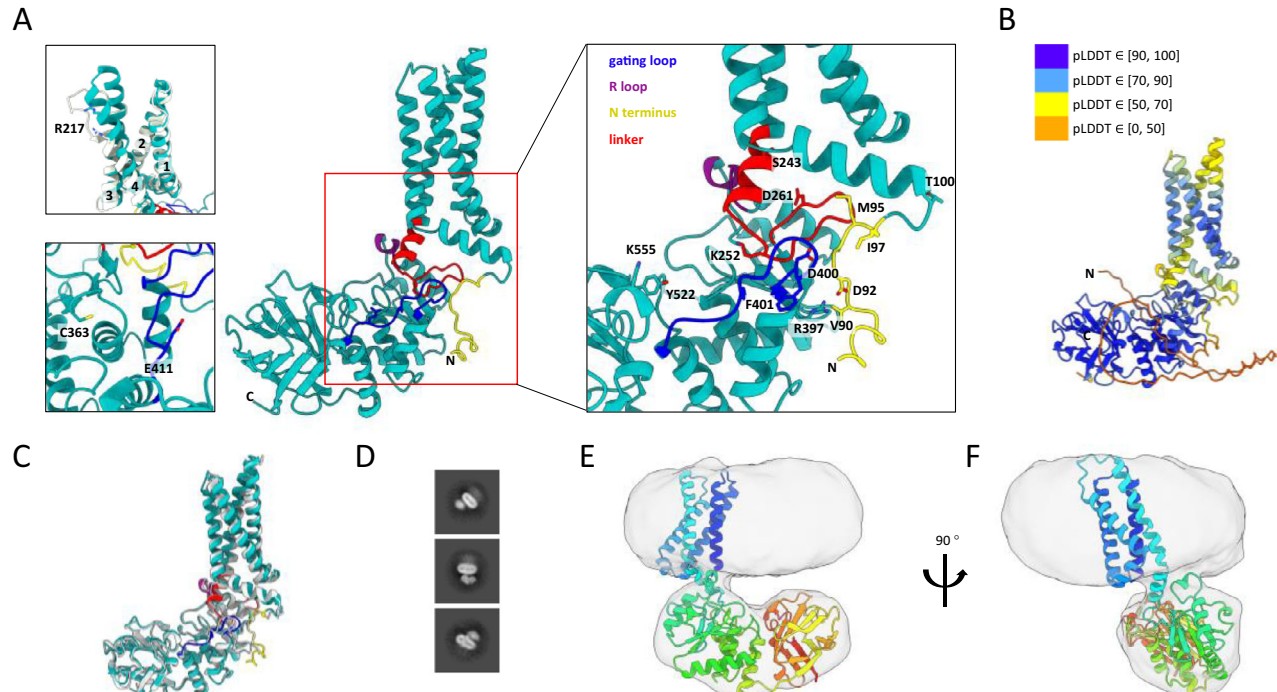

**Fig. 3 | AlphaFold model of Ci-VSP and cryo-EM structure of Dr-VSP in lipid nanodiscs. A** AlphaFold predicted structure of Ci-VSP (with only part of the N-terminus shown). (Inset top left) Superposition of predicted model (cyan) and the crystal structure of Ci-VSP VSD R217E (PDB ID: 4G7V[14]) (white). (Inset bottom left) Zoom in view of the catalytic site with side chain of catalytic residue C363 and gating loop residue 411 shown (top view). (Inset right) Zoom in view of areas with possible interactions between VSD and PD: PD linker (red), R loop (purple), gating loop (blue), and N-terminus (yellow). Important residues were shown.

**B** The predicted structure colored by model confidence score pLDDT from low confidence (orange) to high confidence (blue). **C** Superposition of AlphaFold-predicted models of Ci-VSP (cyan) and Dr-VSP (gray). RMSD value for superpose is 0.889 Å. Same segments are colored with the same scheme as in (**A**). **D** Selected 2D class averages of Dr-VSP in lipid nanodiscs. **E, F** 6.8 Å cryo-EM map (gray) docked with AlphaFold model of Dr-VSP, viewed from the plane of the membrane. Dr-VSP with rainbow colored from N-terminus (blue) to C-terminus (red).

superposed closely with that of Ci-VSP (RMSD: 0.889 Å) (Fig. 3C). We therefore incorporated Dr-VSP (C302S) into nanodiscs for the study with cryo-EM (Supplementary Fig. S7A–D). Two classes of particles were obtained (Supplementary Fig. S7E). One class of particles yielded a structure of the cytoplasmic domain in a dimeric form at a resolution of 3.0 Å (Supplementary Fig. S8A–D; Supplemental Table 1). This structure resembled prior X-ray crystal structures of the isolated Ci-VSP cytoplasmic domain[15], with the E350 gate (E411 in Ci-VSP) occluding access to the active site (Supplementary Movie-3). This conformation suggests the protein was in the resting/inactive state, which Dr-VSP is expected to occupy in the absence of a membrane electric field[8]. The dimeric interface in Dr-VSP buries 744.5 Å² of surface area and involves 21 residues from each monomer, among which 4 hydrogen bonds (E251/G253, G253/E251, Y261/E461, and E461/Y261) and 5 hydrophobic interactions were observed. A similar, albeit smaller, interface is observed in Ci-VSP crystal structures. While diffuse density was observed in the expected position of the nanodisc and TM region (Supplementary Movie-4), these features were not well resolved in the final map, suggesting flexible attachment in the resting/inactive state.

The second class of particles yielded a monomer structure in a nanodisc membrane, with a final resolution of 6.8 Å (Fig. 3D–F, and Supplementary Figs. S7E and S8E, F). The VSD was not resolved, but diffuse density connecting the membrane to the PD suggested that the PD lies immediately "beneath" the VSD. The AlphaFold model of Dr-VSP docked well into the low-resolution monomer structure (Fig. 3E, F), supporting further exploration on the VSD-PD interactions predicted by the AlphaFold model. We therefore explored the functional relevance of the predicted interaction between the N-terminal domain and moving parts of the PD.

## N-terminal domain: voltage-dependent rearrangement and role in coupling

In order to explore the possible role of the N-terminal domain in VSP function, we first tested whether it underwent voltage-dependent rearrangement by incorporating ANAP individually at seven locations in the N-terminal domain of Ci-VSP: W54, E66, L81, V90, M95, I97 and T100. Voltage-dependent ΔFs were barely detected in the predicted disordered section of the N-terminal domain at W54 and E66 but were substantial in the predicted structured section at L81-T100 (Fig. 4A). Interestingly, as seen earlier at C2 domain residue 555[16], the ANAP ΔF at V90, I97 and T100 show two distinct phases of fluorescence change at voltages more positive than 80 mV but only a single phase at less positive ones, indicating sequential conformational changes (Fig. S9). At two of these sites (I97 and T100), the voltage dependence of these phases of N-terminus rearrangement coincided with the F2 and F3 components of S4 motion (Fig. 2B). If the N-terminal domain participates in the voltage-driven rearrangements of the intracellular domain of Ci-VSP, then mutations that affect enzyme activation would be expected to impact N-terminus movement. We tested this by measuring the effect on the T100-ANAP ΔF of mutation K252E in the linker and D400R in the gating loop, as well as L284Q and F285Q in the R-loop, which have been implicated in the late phase of Ci-VSP activation[16]. Each of these mutations dramatically flipped the direction of the ΔF$_{420-460nm}$ (Fig. 4B, C).

To further explore this, we examined the complementary prediction that mutations of the N-terminus should affect the voltage-dependent conformational changes of the linker, gating loop and catalytic site. Deletion of the Ci-VSP N-terminus from residue 2 to midway into the pre-S1 helix (Δ2-105) dramatically reduced the amplitude of the ΔF and the spectral shift ΔR at each of the PD sites except at the

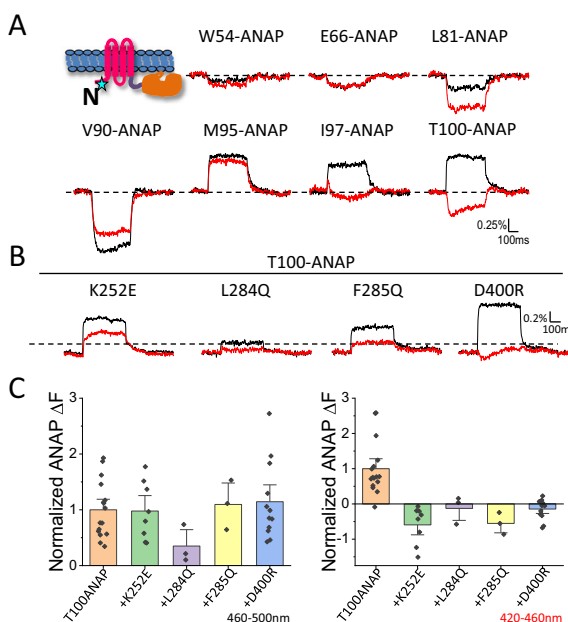

**Fig. 4 | Voltage-driven conformational change in N-terminal domain and perturbation by mutations in linker, R loop and gating loop suggest role for N-terminus in VSP activation. A** Labeling scheme and representative fluorescence traces of ANAP incorporated at different N-terminus sites as shown at 200 mV. Fluorescence was simultaneously detected from 460–500 nm (black) and 420–460 nm (red). **B** Representative fluorescence traces of ANAP incorporated at N-terminus residue 100 with mutation K252E, L284Q, F285Q and D400R at 200 mV. Fluorescence was simultaneously detected from 460–500 nm (black) and 420–460 nm (red). **C** Comparison of ANAP ΔF from residue 100 and from that with mutation K252E, L284Q, F285Q and D400R recorded from 460–500 nm (left) and 420–460 nm (right) (mean ± s.e.m.; $n = 16, 8, 3, 3, 12$ from left to right respectively). Each n represents an independent measurement from an individual oocyte.

S4-proximal linker position 243 (Fig. 5 and Supplementary Fig. S10A, B). The deletion did not change the amount of ANAP-incorporated protein on the membrane, as judged by the level of TMRM labeling of G214C at the outer end of S4 (Fig. S10C), showing that the reduction in ΔF was not due to reduced expression but to altered conformational rearrangements. Thus, there is a reciprocal relationship between the N-terminal domain, on one hand, and the distal linker, gating loop and catalytic site, on the other, in which mutation of the N-terminal domain affects the rearrangements of these other sites and vice-versa.

We next asked whether the N-terminus deletion would also affect functional coupling between the VSD and PD. To test this, we measured the conformational change of S4 in Δ2-105 using TMRM labeling on Q208C. We found that Δ2-105 eliminated the F3 component of Q208C-TMRM ΔF (Fig. 6A, E), as we observed with the VSD-PD uncoupling linker mutant K252E and the PD deletion mutant ΔPD (Fig. 2A). Deleting the N-terminus also had a profound effect on PIP₂→PIP phosphatase activity, eliminating the removal of the 3′ phosphate from PI(3,4)P₂ (the late decrease phase of F-TAPP ΔFRET) and severely attenuating the removal of the 5′ phosphate from PI(4,5)P₂ (the late decrease phase of F-PLC ΔFRET) (Fig. 7), again as seen with linker mutant K252E (Fig. 2). However, unlike K252E, Δ2-105 retained a robust and fast component of 5′ phosphate dephosphorylation from PI(3,4,5)P₃ (the early increase phase of F-TAPP ΔFRET due to increased PI(3,4)P₂) (Fig. 7). These observations suggest that the N-terminal domain plays an important role in VSD-PD coupling, particularly in PIP₂ dephosphorylation.

Using the reverse coupling assay, we tested a series of N-terminus deletions and point mutations to narrow down the portion of the N-terminus that is responsible for its functional effects. Whereas Δ2-53

and Δ54-79 had no effect on the Q208C-TMRM F-V, Δ80-105 eliminated the Q208C-TMRM F3 component, similar to that seen with Δ2-105 (Fig. 6B, E). This suggests that the 26 amino acid stretch from residue 80 to 105, including part of the pre-S1 helix, is the critical portion of the N-terminal domain. This 26-amino acid stretch includes six residues (V90, D91, D92, G93, R94 and M95) that the AlphaFold model situates within interaction distance to a conserved stretch of the distal linker and the beginning of the gating loop (Fig. 3A). To search for potential interactions, we focused on the charged residues, including the two conserved aspartates D91 and D92. Simultaneously neutralizing these two residues with mutation to asparagine (D91N/D92N) reduced the F3 component of the Q208C-TMRM F-V (Fig. 6C). In contrast, mutating two other nearby conserved glutamates to asparagine (E85N/E87N) or a nearby conserved basic residue 94 to glutamine (R94Q) did not reduce the F3 component of the Q208C-TMRM F-V (Fig. 6C). Mutation of D91 and D92 to alanine (D91A/D92A) produced an even more severe disruption of F3, which was similar to that induced by Δ80-105 and Δ2-105.

These observations show that mutations of D91 and D92 affect the voltage sensing rearrangement of the VSD and suggest that they are important to the N-terminus's role in VSD-PD coupling. To test this hypothesis, we studied the impact of D91A/D92A on the voltage-driven conformational changes of the linker, gating loop, catalytic site and C2 domain using ANAP. Similar amounts of ANAP-incorporated protein were detected at the membrane with and without the D91A/D92A mutation, indicating that the double mutant did not affect expression (Supplementary Fig. S10C). As with the larger deletion Δ2-105, although to a smaller extent, mutation of D91A/D92A significantly reduced the amplitude of ANAP ΔF and its spectral shift at all testing sites except in the proximal linker (Fig. 5 and Supplementary Fig. S10A, B).

We then examined the impact of the D91 and D92 mutations on enzymatic function, after confirming that these N-terminal domain mutations did not affect expression (Supplementary Fig. S11). As observed with the large N-terminus deletion, Δ2-105, the D91N/D92N and D91A/D92A double mutants selectively reduced the late decrease phase of F-TAPP ΔFRET (Fig. 7). As with the Q208C-TMRM F-V, the effect on enzymatic activity of D91A/D92A was more severe than D91N/D92N and each of the residues contributed to this effect (Supplementary Fig. S12). As the AlphaFold model predicted proximity between N-terminus residue D92 and residue R397 at the beginning of the gating loop, we mutated R397 to either asparagine or aspartate to neutralize or flip the charge, respectively. The F3 component of the Q208C-TMRM F-V was eliminated in both R397N and R397D (Fig. 6D). Moreover, R397N diminished PI(3,4)P₂→PI(4)P activity while retaining PI(3,4,5)P₃→PI(3,4)P₂ activity, as detected by F-TAPP, and severely reduced PI(4,5)P₂→PI(4)P activity, as detected by F-PLC (Fig. 7A, C). Stronger impact on enzymatic activity was observed with R397D (Fig. 7A, C). These results are consistent with electrostatic interaction between R397 and D91/D92 and suggest that the membrane-proximal N-terminus of the VSD participates in an assembly with the distal VSD-PD linker and gating loop to translate voltage-driven motions in the VSD into changes in phosphatase activity.

## Discussion

VSP is composed of a 4-helix membrane-spanning voltage sensor domain (VSD), which resembles that of voltage-gated cation channels, and a cytosolic PTEN-like lipid phosphatase domain (PD)[1]. An arginine-rich S4 transmembrane segment in the VSD operates as the enzyme voltage sensor. Membrane depolarization drives a rearrangement of S4 that turns on catalytic activity in the PD to dephosphorylate PIPs in the inner leaflet of the membrane. While gating current measurement and voltage clamp fluorometry have provided evidence for S4 motion[9,11,12,24], and protein motion was reported at the N terminal end of the gating loop and in the C2 domain[20], evidence has been missing

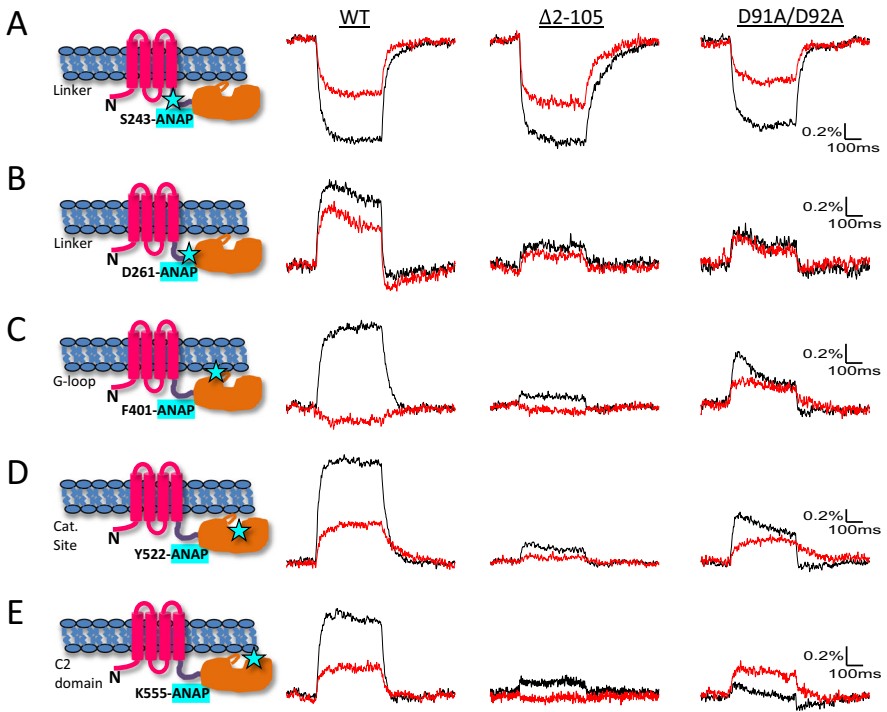

**Fig. 5 | N-terminus deletion and D91A/D92A disrupts conformational coupling.**
**A–E** Representative fluorescence traces from ANAP incorporated at proximal linker 243 (**A**), distal linker 261 (**B**), gating loop 401 (**C**), catalytic site 522 (**D**), C2 domain 555 (**E**), and the corresponding sites with N-terminus deletion Δ2-105 and with D91A/D92A mutation at 200 mV. Fluorescence was simultaneously recorded from 460–500 nm (black) and 420–460 nm (red). The cartoon in each panel indicates the individual labeling site.

for motion of other key intracellular components, including the VSD-PD linker, the R loop, the E411 gate (at the C-terminal end of the gating loop) and the catalytic site. Our experiments show that each of these segments undergo voltage-dependent conformational rearrangements in the full-length protein in the live cell membrane.

Rearrangements in the gating loop and catalytic site were impacted by mutation of residues that have been implicated in the linker-gating loop interaction, which connects voltage-sensing to enzymatic activity ("forward coupling") and transmits the influence of mutations in the catalytic site to S4 ("reverse coupling")[15]. These observations support the gated-access model, in which S4 motion induces a conformational change in the linker and, therefore, in the gating loop, and that turns on catalysis. However, the stronger effects of K252E/Q compared to that of D400R suggest that K252 may have additional interactions. Moreover, neither protein motion nor enzymatic activity was completely disrupted by these mutations. Similarly, Ci-VSP still retained significant enzymatic activity even with a mutation of the key residue that is proposed in the PD displacement model to interact with S4-linker to bring the catalytic site closer to the PIPs in the membrane[16,17]. Together these observations suggest that additional determinants contribute to VSD-PD coupling.

Cryo-EM on Dr-VSP yielded both a high-resolution dimer of the cytoplasmic domain and a low-resolution monomer structure in a nanodisc, which provide structural evidence that a full-length VSP can exist in both monomer and dimer forms. This supports prior findings that, at low density in live cell membranes, Ci-VSP is a monomer[12], whereas at high density it forms dimer[26]. The high-resolution dimer structure reveals residues and interactions that are involved in the dimer formation of the cytoplasmic domain. The dimer structure closely resembles the X-ray crystal structures of the isolated cytoplasmic domain of Ci-VSP, particularly the 3V0F conformation in which the E411 gate occupies the active site pocket[15].

The low-resolution monomer structure, docked plausibly with the AlphaFold model of Dr-VSP, suggests a structural orientation between the VSD and PD, whereby the PD lies immediately beneath the VSD with more inter-domain contact than previously thought. The AlphaFold model predicts that the VSD-PD linker begins as a helix at the end of S4 and then continues as an extended loop, which lies within interaction distance to the gating loop, consistent with the crystal structures[15]. Consistent with a recent disulfide-locking study[17], the AlphaFold linker helix lies in close proximity to the R loop. Although with a lower confidence score, the AlphaFold model also predicts that a section of the N-terminal domain near the pre-S1 helix interacts with both the linker and the gating loop, suggesting that it could serve as a molecular bridge between S4 and the gating loop. In support of this model, our experiments showed that the membrane-proximal portion of the N-terminal domain undergoes voltage-dependent rearrangement. Moreover, mutational analysis showed that the N-terminal domain is essential for voltage-driven rearrangements in the distal linker, gating loop, catalytic site and the C2 domain, as well as for voltage-dependent phosphatase activity. Two residues in the N-terminal domain, D91 and D92, are responsible for much of its functional and conformational impacts and their AlphaFold-predicted electrostatic partner R397, located at the beginning of the gating loop, is similarly critical. In addition to eliminating the late phase of S4 motion, as seen by Q208C-TMRM, mutations in the N-terminal domain—including Δ2-105 and D91/D92—as well as mutation of R397N selectively impact $PIP_2$ to PIP activity while retaining $PIP_3$ to $PIP_2$ activity.

Based on these results, we propose that voltage-driven motion of S4 is transmitted to the catalytic site by a 3-way assembly formed by the distal VSD-PD linker, the PD gating loop and a component, the membrane-proximal N-terminal domain of the VSD (Fig. 8). The core mechanism of voltage sensor to effector coupling is reminiscent of

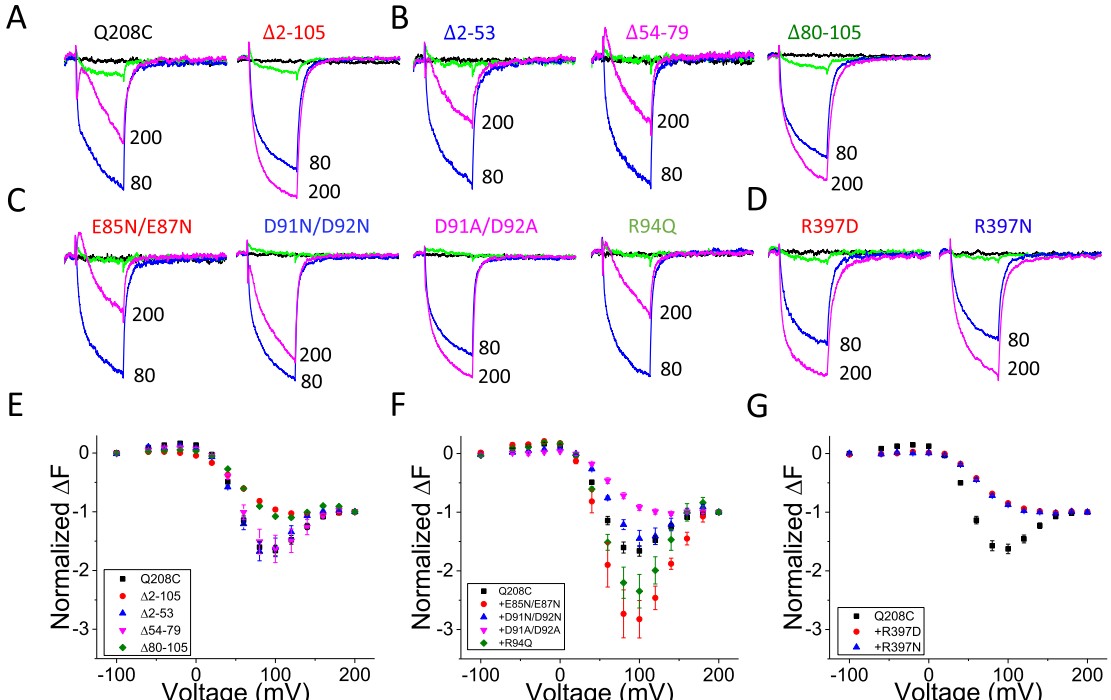

**Fig. 6 | N-terminus mutations and AlphaFold-predicted interacting partner of D91/D92 alter voltage-driven S4 conformational changes. A–C** Representative fluorescence traces from TMRM-labeled Q208C of WT Ci-VSP and different N-terminus mutations as indicated at 200 mV (magenta), 80 mV (blue), 20 mV (green) and −100 mV (black). **D** Representative fluorescence traces from TMRM-labeled Q208C of Ci-VSP with mutation of the gating loop residue R397 (the AlphaFold-predicted interacting partner of D91/D92) at 200 mV (magenta), 80 mV (blue), 20 mV (green) and −100 mV (black). **E–G** Comparison of F-V curves of WT and different N-terminus deletions (**E**) (mean ± s.e.m.; $n = 80, 4, 3, 4, 10$, in the top to bottom order shown in the inset), point mutations (**F**) (mean ± s.e.m.; $n = 80, 4, 4, 16, 3$, in the top to bottom order shown in the inset) and mutations of R397 (**G**) (mean ± s.e.m.; $n = 80, 13, 13$, in the top to bottom order shown in the inset). Fluorescence measured at the end of each test voltage was normalized to that obtained at 200 mV. Each n represents an independent measurement from an individual oocyte.

how ion flux is controlled by VSDs in voltage-gated $K^+$, $Na^+$ and $Ca^{2+}$ channels. In channels, membrane depolarization drives the S4 into the "up" state, repositioning the post-S4 segment and S5, to which it is connected, to exert force on the internal "gate" formed by S6. In VSP, the outward movement of S4 drives structural rearrangements in the post-S4 distal VSD-PD linker, which, in turn, pulls on the gating loop to open access to the catalytic site. A separate interaction—between the membrane-proximal N-terminal domain and residue R397 in the gating loop—is associated with the transition in substrate preference from $PIP_3$ to $PIP_2$ at larger depolarizations. Neighboring residue R398, which forms hydrogen bonds with the active site loop[15], may provide the route for the N-terminal domain to impact substrate selection. In this way, voltage-enzyme coupling in VSPs sequentially opens access to the catalytic site and modifies the catalytic site to switch substrate preference via a 3-way assembly of the distal VSD-PD linker, the PD gating loop and the membrane-proximal N-terminal domain of the VSD. The severe impact of the linker/gating loop mutations on VSP function is consistent with the gating access model, although further studies are needed to confirm that gating loop conformational changes alter catalytic site access. The persistence of some voltage-dependent structural rearrangements and enzyme activity in the D400 mutant may reflect other paths of VSD-PD coupling such as proposed in the PD displacement model.

## Methods

### Ethical statement

This study complies with all relevant ethical regulations approved by the Animal Care and Use Committee (ACUC) of University of California, Berkeley (protocol ID: AUP-2015-04-7522-3).

### Constructs and RNAs

Plasmid pANAP was a gift from Peter Schultz (Addgene plasmid # 48696; http://n2t.net/addgene:48696; RRID:Addgene_48696). The plasmid encoding for WT Ci-VSP in pSD64TF vector was kindly provided by Y. Okamura (Osaka University). The Kir2.1 construct was kindly provided by E. Reuveny (Weizmann Institute of Science). Various mutants were made in the same vector using quick change site directed mutagenesis. The construct encoding for F-PLC and F-TAPP was originally generated in the pGEMHE vector by ligating the sequence encoding for GRP1-PH (provided by M. Matsuda, Kyoto University) with PLCδ1-PH (provided by T. Meyer, Stanford University) and TAPP1-PH (provided by T. Balla, NIH), respectively. Ci-VSP constructs contain G214C in ANAP experiments and Q208C in TMRM and enzyme activity experiments. To make RNA for oocyte injection, cDNA of Ci-VSP were linearized with XbaI and in-vitro transcribed with SP6 mMachine kit. The cDNA of F-PLC and F-TAPP was linearized with NheI and in-vitro transcribed with T7 kit. Full-length Dr-VSP construct was subcloned into a modified pPICZ-B vector (Life Technologies, Inc.), fused with a human rhinovirus 3C protease-cleavable C-terminal eGFP-10× histidine.

### Unnatural amino acid incorporation

After *Xenopus laevis* oocytes were harvested, the outer follicular cell layer was removed by collagenase digestion. Plasmid pANAP (~0.2 μg/μl) was injected into the nuclear of the oocytes. After 24 hr, mRNA of Ci-VSP constructs with various engineered TAG stop sites (1.0 μg/μl) was mixed with ANAP (2 mM) (Cayman Chemical) at 1:1 ratio and injected into the cytosol of the oocyte. The injected oocytes were incubated at 16 °C for 48 hr before the recording. Each Ci-VSP

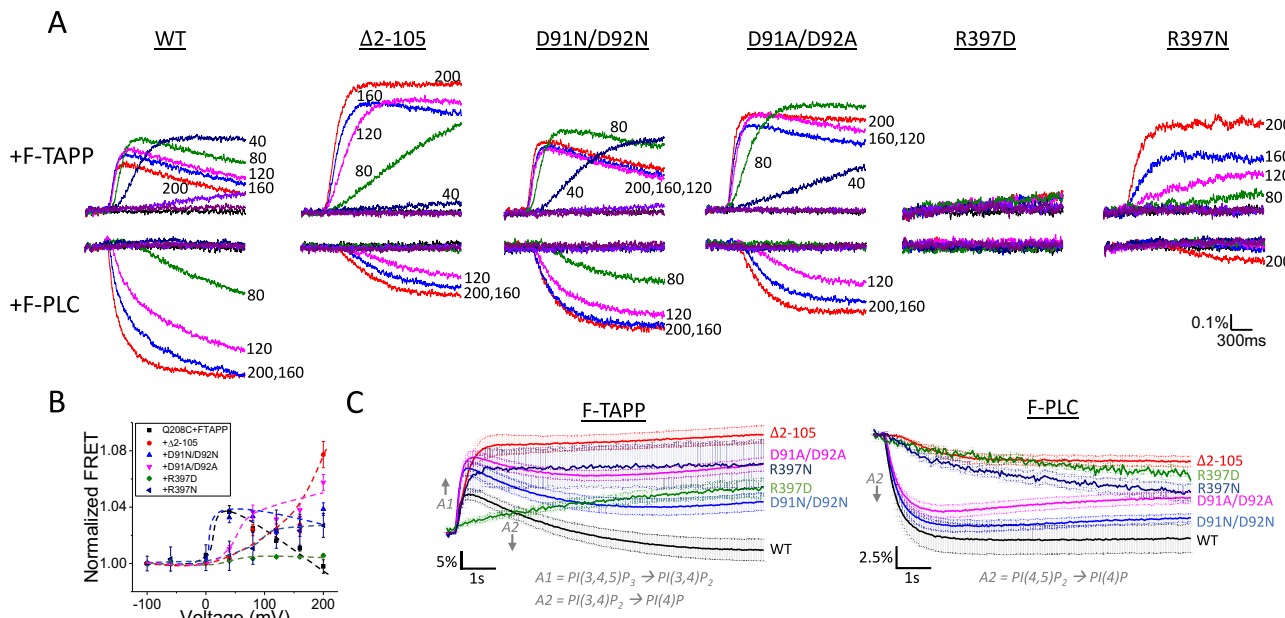

**Fig. 7 | N-terminus mutations that alter conformational coupling and R397D/N also affect enzyme activity. A** Representative fluorescence traces from F-TAPP (top row) or F-PLC (bottom row) co-expressed with WT, Δ2-105, D91N/D92N, D91A/D92A, R397D and R397N mutation using the 2-s test protocol (See "Methods"). Some test voltages are shown next to corresponding traces. **B** Comparison of the voltage dependence of FRET changes from F-TAPP when co-expressed with different Ci-VSP constructs as indicated. FRET values were measured at the end of the 2-s pulse to each voltage and normalized to that of −100 mV ($n$ = 11, 9, 10, 10, 11, 11 in

the listed order respectively). **C** Averaged fluorescence traces from F-TAPP and F-PLC co-expressed with different Ci-VSP constructs (as indicated and color-coded) using the long (15 s) test protocol. Traces from multiple rounds of experiments were averaged and experiments on WT and mutants were performed on the same batch of oocytes for each round (mean ± s.e.m.; $n$ = 9, 8, 8, 10, 5, 12 (for FPLC) and $n$ = 14, 12, 12, 15, 10, 7 (for FTAPP) for Q208C, Δ2-105, D91N/D92N, D91A/D92A, R397D and R397N, respectively). Each n represents an independent measurement from an individual oocyte.

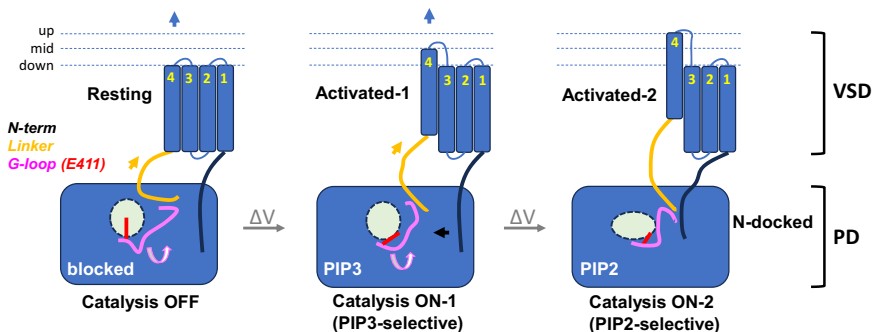

**Fig. 8 | Model of voltage-catalysis coupling in VSPs.** A 3-way interaction between the distal VSD-PD linker (orange), the PD gating loop (pink) and the S1-proximal N-terminal domain (black) couples two sequential depolarization-driven outward movements of S4 to rearrangements in the PD that toggle it between three functional states. In the first step, S4 movement reorients the VSD-PD linker and the

interacting gating loop to swing glutamate 411 (red) away from the catalytic pocket, thereby opening access for entry of $PIP_3$ substrate. In the second step, further reorientation of the linker-gating loop engages the membrane-proximal N-terminal domain of the VSD and alters the catalytic site to switch substrate preference to $PIP_2$.

construct for the ANAP experiment includes the G214C mutation, which allows the identified ANAP-incorporated oocytes to be labeled with TMRM to assess expression on the membrane.

### Voltage-clamp fluorometry
Oocytes were injected with 50 nl of Ci-VSP mRNA at 1.0 µg/µl (for TMRM labeling experiments) or Ci-VSP mRNA mixed with F-PLC or F-TAPP mRNA at 1:2 ratio (for enzyme activity assay with these FRET reporters). The oocytes were then incubated in ND-96 solution (96 mM NaCl, 2 mM KCl, 1.8 mM CaCl₂, 1 mM MgCl₂, 50 mg/ml gentamicin, 2.5 mM Na-pyruvate and 5 mM HEPES, pH 7.5) for 36−48 hr at 16 °C. The TMRM labeling experiments were done similarly as previously described[9,24]. Briefly, the injected oocytes were incubated with 25 µM tetramethylrhodamine-6-maleimide (Invitrogen) on ice for 1 hr,

washed thoroughly with ND-96 and stored at 12 °C in the dark till testing.

To record fluorescence changes from TMRM, oocytes were held at −100 mV and then pulsed to a test voltage (from −60 mV to 200 mV in 20 mV increments) for 1.2 s every 5 s. Two protocols were used in ANAP experiments. Protocol 1 was used to generate F-V curves and was similar to the TMRM voltage protocol except that the test pulse was 220 ms every 2 s. The whole series was repeated 3 times and ANAP traces at each voltage were averaged. Protocol 2 included a pulse from −100 mV to a test voltage (200 mV, 160 mV, 120 mV, 80 mV, 40 mV or 0 mV) for 400 ms and the same test pulse was repeated 8 times with 2 s between repeats and more than 15 s between each test pulses. All eight repeats were averaged to generate the final trace. To measure the enzyme activity with the F-PLC and F-TAPP reporters, a short or a long

protocol was used. For the short protocol, oocytes were pulsed from −100 mV to different test voltages (from 200 mV to 0 mV in 40 mV decrements and to −60 mV and −100 mV) for 2 s every 60 s (short protocol). For the long protocol, oocytes were pulsed from −100 mV to 200 mV for 15 or 20 s. Each oocyte was only subjected to one round of either the short or the long protocol. Shutter was always off in between pulses to reduce photobleaching.

Two-electrode voltage clamp fluorometry was performed with an Olympus IX-71 inverted fluorescence microscope equipped with a 20 × 0.75 NA objective (Olympus UApo/340), 100-W mercury arc lamp (Hamamatsu), Uniblitz shutter (Vincent Associates), V-1585 Oscilloscope (Hitachi), Dagan CA-1B amplifier, Digidata-1440A digitizer (Molecular Devices) and pClamp 10 software (Molecular Devices). Signals were detected with either a HC120-05 Photomultiplier (Hamamatsu) or a pair of PMT-100 Photomultipliers (Applied Scientific Instrumentation) attached to a Photoport Bean Splitter (Applied Scientific Instrumentation). For the TMRM labeling experiments, a Chroma filter cube containing an excitation filter (HQ 535 nm/50 nm), dichroic (Q 565 nm-LP), and an emission filter (HQ 620 nm/60 nm) was used. For the ANAP incorporation experiments, light was filtered through an excitation filter (AT 350 nm/50 nm) and a dichroic (T 400 nm lp), and the signal was split by a dichroic (ZT458rdc-UF1) to two emission filters ET440 nm/40 nm, ET480 nm/40 nm (all from Chroma Technology). For the F-PLC and F-TAPP experiments, excitation filter (ET 420 nm/40 nm), dichroic (440dcrxu), beam-splitting dichroic (495dcsp), two emission filters (HQ 470 nm/20 nm) and (ET 535 nm/30 nm) were used (all from Chroma Technology). All signals were low-pass filtered at 1 kHz using an eight-pole Bessel filter (Dagan).

### Electrophysiological measure of activity

For experiments in which the catalytic activity of Ci-VSP was measured through the activation of the Kir2.1 R228Q channel (Kir2.1Q), the oocytes were first injected with pANAP, various Ci-VSP mRNA and ANAP as described in the previous section, followed by the injection of 50 nl mRNA of Kir2.1Q (0.3 ug/ul) a day before recording. The R228Q mutation was used to alter the sensitivity of Kir2.1 for PI(4,5)P$_2$ into detectable range[1,12,27]. All injected oocytes were first tested for ANAP incorporation (200 mV for 400 ms). Oocytes with successful ANAP incorporation were then tested for phosphatase activities using protocols described previously[1,9,12]. Briefly, the oocyte was depolarized to +60 mV for 100 s to activate Ci-VSP. It was then hyperpolarized to −100 mV and a brief 50 ms voltage ramp to 50 mV was given every 10 s for 40 times. Steady state currents were measured at −100 mV before each ramp and leak current were measured at +50 mV at the end. Assuming a voltage-independent linear leak, the leak-subtracted current ($I_{ls}$) was calculated as following: $I_{ls} = I_{-100\ mV} + 2I_{+50\ mV}$, and the normalized activity as $\Delta I / I_{max} = (I_{ls\ -100\ mV\ last} - I_{ls\ -100\ mV\ first}) / I_{-100mV\ last}$. The recording solution contained 96 mM KCl, 2 Mm NaCl, 1 mM MgCl$_2$, 1.8 mM CaCl$_2$, 8 mM KOH, 10 mM HEPES, pH 7.4. All other setups were the same as for the voltage-clamp fluorometry.

### Data analysis

The averaging of fluorescence traces, baseline adjustment, FRET calculation, normalization and curve fitting were processed in Clampfit 11 (Molecular Devices), Microsoft Excel and Origin 9 software. The baseline for each voltage trace was corrected for bleaching by measuring the rate of fluorescence decrease before and after the test pulse. The change of fluorescence was measured at the end of each test voltage, and it was either normalized to the maximal test voltage (200 mV for TMRM F-V experiments) or to each ANAP labeling site without extra mutation (for ANAP ΔF amplitude comparison).

Ratiometric analysis of ANAP fluorescence was based on simultaneous measure at two wavelength ranges (420–460 nm and 460–500 nm) at the holding voltage of −100 mV (baseline) and at the end of the depolarizing voltage step to 200 mV (test). Ratio R

was calculated as F$_{460-500\ nm}$ / (F$_{460-500\ nm}$ + F$_{420-460\ nm}$) at holding voltage (R$_{baseline}$) and at testing voltage (R$_{test}$), and ΔR / R$_{baseline}$ = (R$_{test}$ − R$_{baseline}$) / R$_{baseline}$. Therefore, ΔR / R$_{baseline}$ measured the change of fluorescence ratio of two emission channels when VSP are activated at 200 mV.

Statistical analysis was performed as follows. Each set of experiments was repeated for at least two independent rounds on at least seven oocytes harvested from at least two different frogs. Each set of controls and tests was performed and compared on the same batch of oocytes on the same day. Neither randomization nor blind studies were used. All the error bars shown in the figures are standard error of mean (s.e.m.) and sample sizes are reported in the figure legends for individual experiments.

### Protein expression and purification

The protein expression and purification process are similar to what have been described before[28]. In short, the p-PICZ-B-Dr-VSP plasmid was first linearized with PmeI and then was transformed into the P. pastoris (Invitrogen) strain SMD1163 by electroporation. After growing on YPDS plates with Zeocins (0.5 mg/ml), individual colonies were selected for small liquid culture and the protein expression levels were analyzed by fluorescence size-exclusion chromatography (FSEC). The best expressed colony was selected for large-scale expression, which started from a small liquid culture with 0.5 mg/ml Zeocin in BMGY (Buffered Glycerol-complex Medium) overnight. It was then transferred to large BMGY culture with 25 ug/ml Zeocin at 30 °C till OD600 reached around 25. The culture was then pelleted, resuspended and grown in BMMY (Buffered Methanol-complex Medium) at 27 °C for 36–48 hr for protein expression. Yeast cells were pelleted, flash-frozen in liquid nitrogen and stored at −80 °C.

For protein expression, the frozen yeast cells (-50 g) were milled (Retsch model MM301) for five cycles of 3 min at 25 Hz. The powder was then added to pre-chilled 100 ml lysis buffer containing 50 mM Tris pH 8, 150 mM NaCl, 1 mM phenylmethyl-sulfonyl fluoride, 5 mM MgCl$_2$, 5 mM Dithiothreitol (DTT), 10 μl benzonase nuclease (EMD Milipore), 1 mM E64, 1 mg/ml pepstatin A, 10 mg/ml soy trypsin inhibitor, 1 mM benzamidine, 1 mg/ml aprotinin, 1 mg/ml leupeptin. After sonication for 4 min, the lysate was centrifuged at 150,000 g for 45 min. The pellet of the membrane fraction was then homogenized with a Dounce homogenizer in 100 ml pre-chilled extraction buffer (same composition as the lysis buffer but with 1% n-dodecyl-β-d-maltopyranoside (DDM; Anatrace) and 0.2% cholesterol hemisuccinate Tris salt (CHS; Anatrace)). The homogenized membrane fraction was stirred for 2 hr at 4 °C and then centrifuged at 33,000 g for 45 min. The supernatant was then mixed with 3 ml of Sepharose resin that was coupled to the anti-GFP nanobody by gentle stirring for 2 hr at 4 °C. The resin was collected in a column and washed with 100 ml Buffer 1 (20 mM Tris, 500 mM NaCl, 1 mM DTT, 0.025% DDM and 0.005% CHS, pH 8), 30 ml Buffer 2 (20 mM Tris, 150 mM NaCl, 1 mM DTT, 0.025% DDM and 0.005% CHS, pH 8). Human rhinovirus 3 C protease (-0.5 mg) was added into the washed resin in 5 ml Buffer 2 and rocked gently overnight. The Dr-VSP protein cleaved from the GFP was then eluted and concentrated to -0.7 ml with an Amicon Ultra spin concentrator (50 kDa cut-off, MilliporeSigma). The concentrated protein was run in Buffer 2 through a Superose 6 Increase 10/300 GL column (GE Healthcare) on a NGC Medium-Pressure Chromatography Systems (Bio-Rad) controlled by the ChromLab 6 software. The peak fractions were collected and spin concentrated.

### Nanodisc assembly

To prepare for the freshly purified Dr-VSP protein and the Nanodisc assembly, a 2:1:1 DOPE:POPS:POPC lipid mixture (mass ratio; Avanti Polar Lipids) was mixed with Brain PI(3,4)P$_2$ (at relative 0.2 mass ration with the lipids, Avanti Polar Lipids). The mix were then dried under argon, washed with pentane three times and dried under vacuum in

the dark overnight. Dried lipids mixed with PI(4,5)P$_2$ were rehydrated in Buffer 3 containing 20 mM Tris, 150 mM NaCl, 5 mM EGTA, pH 8, and sonicated till it reached clarity. DDM was added to the lipid mix to reach a final molar ratio of 5 DDM:3 Lipid. The Dr-VSP protein was mixed with the lipids and incubated at 4 °C for 30 min, followed by the addition of MSP2N2 protein and 3 hr incubation at 4 °C. The final molar ratio of Dr-VSP:MSP2N2:lipid was 1:4:400. Two 200 mg of Bio-Beads SM2 resin (Bio-Rad) was added, separated by 30 min incubation in between, and the mixture was rotated at 4 °C overnight. Before used, the resin was prepared by sequential washing in methanol, water and Buffer 3 and weighed damp.

After removing the Bio-Beads by centrifugation, Dr-VSP protein incorporated into Nanodisc was purified on a Superose6 increase column run in Buffer 3. The peak fractions were collected and spin concentrated (50 kDa molecular weight cut-off) to 5.6 mg/ml for grid preparation. The peak fractions were run on a 12% SDS-PAGE gel to confirm the present of Dr-VSP and MSP2N2 proteins.

### EM sample preparation and data collection
Dr-VSP in MSP2N2 nanodisc was centrifuged at $21,000 \times g$ for 5 min at 4 °C. A 3.5 μL sample was applied to UltrAuFoil 300 mesh R1.2/1.3 gold grids (Quantifoil, Großlöbichau, Germany) that were freshly glow discharged for 25 s. Sample was incubated for 5 s at 4 °C and 100% humidity prior to blotting with Whatman #1 filter paper for 3 s at blot force 1 and plunge-freezing in liquid ethane cooled by liquid nitrogen using a FEI Mark IV Vitrobot (FEI/Thermo Scientific, USA). Grids were clipped and transferred to a FEI Talos Arctica electron microscope operated at 200 kV. Fifty frame movies were recorded on a Gatan K3 Summit direct electron detector in super-resolution counting mode with pixel size of 0.5575 Å. The electron dose rate was $8.849\,e^-\,Å^2\,s^{-1}$ and the total dose was $50.0\,e^-\,Å^2$. Nine movies were collected around a central hole position with image shift and defocus was varied from $-0.6$ to $-1.8\,\mu m$ through SerialEM.

### Cryo-EM data processing
5037 micrograph movies were motion-correction with dose-weighting using RELION3.1's implementation of MotionCor2 and "binned" 2x from super-resolution to the physical pixel size. CTFFIND-4.1 was then used to estimate the contrast transfer function (CTF). Micrographs with a CTF maximum estimated resolution worse than 4 Å were discarded, yielding 3993 micrographs. Particle images were auto-picked first with RELION3.1's Laplacian-of-Gaussian filter and, following initial clean-up and 2D-classification, templated-based auto-picking was performed, yielding 25,668,338 particles. After importing these particles to CryoSPARC, iterative 2D classification, ab initio reconstruction (3 classes), heterogeneous refinement yielded 3 classes. The first class shows PD domain clearly, the second class shows bad particles, and the third class shows nanodisc clearly. For the first class, particles were used for training in Topaz, and the resulting Topaz model used to repick 1,334,220 particles. Iterative 2D classification, ab initio reconstruction, heterogeneous refinement, and non-uniform refinement in CryoSPARC yielded 208,690 particles. For the third class, particles also were used for training in Topaz, and the resulting Topaz model used to repick 1,555,437 particles. Iterative 2D classification, ab initio reconstruction, heterogeneous refinement, and non-uniform refinement in CryoSPARC yielded 46,944 particles. UCSF pyem was used for conversion of files from cryoSPARC to RELION formats.

### Model building and refinement
The final map of the first class was used for PD domain modeling. The AlphaFold2 PD domain of Dr-VSP model was rigid body fit to the density in ChimeraX and used as a foundation for manual model building in Coot. The model was real space refined in Phenix and assessed for proper stereochemistry and geometry using Molprobity

The final map of the third class was used for Dr-VSP monomer modeling. The AlphaFold2 Dr-VSP model was docking to the density in Phenix. Structures were analyzed and figures were prepared with ChimeraX and Adobe Illustrator software.

### Reporting summary
Further information on research design is available in the Nature Portfolio Reporting Summary linked to this article.

## Data availability
The data that support this study are available from the corresponding authors upon request. The final cryo-EM map generated in this study has been deposited in the Electron Microscopy Data Bank under accession code EMD-45178 (*Danio rerio* VSP C302S dimer reconstituted in MSP2N2 nanodiscs). The final atomic coordinates have been deposited in the Protein Data Bank (PDB) under accession codes 9C49 (*Danio rerio* VSP C302S dimer reconstituted in MSP2N2 nanodiscs. The crystal structure of Ci-VSP VSD R217E (4G7V) and cytosolic domain (243-576, C363S, form II) was referred to in the study (3V0F). The source data underlying Figs. 1I–J, 2B, D, E, 4C, 6E–G, 7B, C and Supplementary Figs. S1, S2C, S3A, B, S5, S9, S10, S11, S12C are provided as a Source Data file. Source data are provided with this paper.

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

## Acknowledgements

We thank Shimon Yudovich for helpful advice and discussions on the ratiometric analysis of ANAP fluorescence spectra. We thank all other members of the Isacoff and Brohawn labs for helpful comments. Support was provided by the National Institutes of Health (1F32NS101816 to Y.Y, R01GM117051 to E.Y.I. and GM145869 to S.B.) and New York Stem Cell Foundation (to S.G.B.). E.Y.I. is a Weill Neurohub Investigator.

## Author contributions

Y.Y. and E.Y.I conceived the project. Y.Y. performed the fluorescence and electrophysiology experiments and analysis under the guidance of E.Y.I. Y.Y., L.Z., and B.L. performed the protein purification. L.Z. and B.L. performed the cryo-EM imaging and analysis under the guidance of S.B. L.Z. and Y.Y. performed the structural modeling. Z.F. helped with the molecular biology. Y.Y., S.B., and E.Y.I. wrote the manuscript with input from all of the authors.

## Competing interests

The authors declare no competing interest.
