## [Peer Review File · Nature Communications]

Coupling sensor to enzyme in the voltage sensing phosphataseREVIEWER COMMENTS

Reviewer #1 (Remarks to the Author):

This study uses an impressive combination of biophysical tools to provide new insights into the mechanism of coupling between membrane voltage and enzymatic activity of the voltage-sensitive phosphatase (VSP). Besides the physiological importance of the enzyme, this work will be of considerable interest for the scientists in the ion channel field, because of the unique structure of this enzyme, where a voltage sensor domain (VSD) common to all voltage-gated channels is coupled to a cytosolic phosphatase domain (PD) rather than to a channel pore. The mechanism of coupling between the VSD and PD has been the subject of extensive work of several research groups, Isacoff's included, but remains unclear and there are several controversies that need to be resolved. The new study provides novel insights and details that have been missing or controversial. The main new findings are mostly related to the discovery of an important role for the previously unexplored cytosolic N-terminal domain (NTD), and details of dynamics of movement of key parts of the PD. Interactions between NTD and key elements of PD have been proposed by an AlphaFold structural model and supported by novel cryo-EM data, extensive mutagenesis, resolution of movements of S4 using a standard TMRM label and of different cytosolic parts using a fluorescent artificial amino acid ANAP, and functional assays: electrophysiological (inhibition of a Kir2.1 channel variant) and, importantly, with two FRET reporters of PI(4,5)P2 and PI(3,4)P2 levels. The reporters allowed an accurate, time-resolved analysis of the kinetics of dephosphorylation (by the activated VSD) of PIP3 to PI(4,5)P2 and PI(3,4)P2, and then of the PIP2s themselves. A very interesting and important conclusion from the combined approach is that NTD specifically regulates substrate selectivity (PIP3 vs. PIP2); the dephosphorylation of PIP2s (the second step) can be selectively eliminated by removing NTD a.a. 91/92 and the interacting residues R397 from the PD. Overall, this is an important study with a significant element of novelty, that resolves a previously incompletely understood mechanism. The paper is lucidly written and the illustrations are clear and well explained. I have no major critique; here are several questions and suggestions.

1. Fig. 2 C-E: have you tried the Δ PD construct as the negative control (should eliminate all FRET)?
2. I cannot comment much on cryo-EM results, but admittedly the resolution is quite low, especially for the TMD; a brief discussion on the limitations would be helpful.
3. Just a suggestion for another functional test of Δ NT(2-105) or the double D91A/D92A mutant: if PIP2 dephosphorylation is absent, the level of PIP2s will at least temporarily rise following the activation of these VSP mutants. Is it possible to use a functional reporter, such a highly PIP2-sensitive channel, to strengthen this conclusion? A temporary enhancement of channel's activity may be expected. Kir2.1Q (R228Q) has an intermediate-small basal current compared to WT Kir1.2 (Zhang et al. 1999) and may be bidirectionally sensitive to PIP2 levels. Alternatively, a partly G β γ -activated GIRK channel may be responding to an increase in PIP2 in the membrane.
4. In the introduction, a clear distinction is made between the gated access model and the PD displacement mode of VSD activation. In the Discussion, it might be helpful to better clarify to the reader whether the findings of this work support/extend one of these models, or both (a synthesis?).

Minor points

In Fig. 3 (the AlphaFold2 model) it would be nice to have R397 and the opposing D91/D92 labeled.

In Figures 1, 2, and others: in Y axis, Δ shows as \square ; same in Fig. 2 for Δ PD.

p. 8, sentence starting with "While diffuse density..." has a different font color (dark gray).

Methods, p. 14: "the vitelline membrane was removed by collagenase digestion" is incorrect. Collagenase treatment removes outer epithelial and follicular cell layers, leaving the vitelline membrane intact.

Reviewer #2 (Remarks to the Author):

Dear Editor,

This extensive study by Yawei Yu and Collaborators investigates the mechanism of the operation of voltage-sensing phosphatases (VSP). These enzymes, which possess a transmembrane voltage-sensing domain (VSD) coupled to a cytosolic lipid phosphatase domain (PD), catalyze the dephosphorylation of membrane phosphoinositol phosphates (PIPs) in response to cell depolarization.

The work addresses the fundamental question of how conformational rearrangements of the voltage-sensing domain propagate to the catalytic site of *Ciona intestinalis* VSP (CiVSP), and attempts to identify the cytoplasmic structures involved in the voltage dependent process.

The experimental approach is powerful and combines Voltage Clamp Fluorometry (VCF, a technique pioneered by this group) of Ci-VSP, CryoEM of *Danio rerio* VSP (Dr-VSP) into lipid nanodiscs, FRET-based assessment of enzymatic function, and structural modeling.

By tracking the wave of conformational changes downstream the voltage sensing domain using VCF, the study revealed significant structural rearrangements in three regions: the distal VSD-PD linker, the PD gating loop and the N-terminal domain of the VSD. Interesting CryoEM studies complement the fluorometry information revealing a dimeric structure of the cytoplasmic domain at 3-angstrom resolution and a full-length Dr-VSP at lower resolution.

Abundant experimental and computational work supports a refined model of Voltage-catalysis coupling in VSP. A two-step voltage sensor activation triggers: 1) a reorientation of the VSD-PD linker and the interacting gating loop to open the catalytic for PIP3 and 2) further reorientation of the linker-gating loop and N-terminal domain of the VSD to switch preference for PIP2 at the catalytic site.

Overall, this study advances the field by providing a comprehensive view of the molecular events responsible for the voltage dependence of VSP catalytic activity.

There are some points that need to be addressed or require further clarifications.

Main points:

1) How do the Authors interpret the difference in V-dependence reported by ANAP at positions shown in Fig. S1? Line 139: "Compared to labeling sites 214 in S4 and 243 at the beginning of the linker, the voltage dependence of dye fluorescence (F-V) was shifted positively in the distal linker, gating loop and catalytic site (Fig. S1), suggesting that outward movement of S4 drives subsequent rearrangements that turn on enzyme activity."

Why does a "positive" shift suggest that S4 drives subsequent rearrangements that turn on enzyme activity?

Could the voltage shift be a manifestation of a cooperative step taking place at the PD? I am asking because CryoEM data have shown that full-length VSP forms dimers.

2) The collection of ANAP fluorescence at two different wavelengths is a powerful experimental tool: can ANAP spectral changes (as captured in Fig. 1 J) be interpreted more mechanistically also in view of the structural data from this and previous work? The value of the extensive experimental characterization is sometime lost, such as in the sentence "The reductions in ΔF amplitude and the shifts of ANAP spectrum suggest that the mutations changed either the local environment of the labeling sites or the nature of the rearrangement."

3) The co-expression experiments of Ci-VSP and Kir2.1 are good controls to probe the phosphatase activity of ANAP-substituted Ci-VSP constructs (Fig S2). They show that D261- and G409-ANAP

enzymatic activity is practically abolished. I am wondering how the results from VCF experiments using these two construct should be interpreted. Please explain.

4) Line 133: "ANAP deltaFs observed from residue 401, near the beginning of the gating loop (Fig. 1E), and a C2 domain residue 555 (Fig. 1H) the homolog of which was suggested to associate with the membrane in PTEN23, were consistent with a previous study." Please explain how these VCF data are "consistent with a previous study."

5) I did not find methods for statistical analysis (error bars definition, number of replicates, etc.)

Other points:

1) Line 223: "Besides the high sequence identity with Ci-VSP (48%) (Fig. S6), Danio rerio VSP (Dr-VSP) has an AlphaFold model superposed closely with that of Ci-VSP (RMSD: 0.889 Å) (Fig. 3C), but Dr-VSP with the active site cysteine mutated to serine (C302S) showed better protein expression and stability." This is an unclear sentence: consider revising it.

2) Fig 8: this figure shows a background reticle of dotted lines. They seem unnecessary.

3) Fig S1: is missing the x axis.

4) Fig S2: Kir recordings have not scale bar (but they do not seem normalized).

5) Line 492: missing "e" in baseline

6) Line 705, Fig 3B: Score for low confidence is orange color, not red as indicated in legend.

7) Because of the large number of labeling positions and mutations tested, some parts of the manuscript are difficult to follow. A simple schematic drawing showing the VSP regions discussed (linkers, R/gating loop, catalytic site), mutations and labeling positions, will help to follow the study from the beginning, before more structural information is revealed (Fig. 3).

We thank the reviewers for the constructive comments. In response to the comments, we have added two new sets of experiments: a control experiment for the analysis of the effect of mutations on enzyme activity with our FRET reporters (new panels to Fig. 2D and E) and an additional analysis of the effects of our large N terminus deletion on enzymatic activity using an ion channel readout (Fig. R1). We have also made modifications to several parts of the paper to address the points raised by the reviewers. These are outlined below in our point-by-point response to the reviews.

Reviewer #1 (Remarks to the Author):

This study uses an impressive combination of biophysical tools to provide new insights into the mechanism of coupling between membrane voltage and enzymatic activity of the voltage-sensitive phosphatase (VSP). Besides the physiological importance of the enzyme, this work will be of considerable interest for the scientists in the ion channel field, because of the unique structure of this enzyme, where a voltage sensor domain (VSD) common to all voltage-gated channels is coupled to a cytosolic phosphatase domain (PD) rather than to a channel pore. The mechanism of coupling between the VSD and PD has been the subject of extensive work of several research groups, Isacoff's included, but remains unclear and there are several controversies that need to be resolved. The new study provides novel insights and details that have been missing or controversial. The main new findings are mostly related to the discovery of an important role for the previously unexplored cytosolic N-terminal domain (NTD), and details of dynamics of movement of key parts of the PD. Interactions between NTD and key elements of PD have been proposed by an AlphaFold structural model and supported by novel cryo-EM data, extensive mutagenesis, resolution of movements of S4 using a standard TMRM label and of different cytosolic parts using a fluorescent artificial amino acid ANAP, and functional assays: electrophysiological (inhibition of a Kir2.1 channel variant) and, importantly, with two FRET reporters of PI(4,5)P2 and PI(3,4)P2 levels. The reporters allowed an accurate, time-resolved analysis of the kinetics of dephosphorylation (by the activated VSD) of PIP3 to PI(4,5)P2 and PI(3,4)P2, and then of the PIP2s themselves. A very interesting and important conclusion from the combined approach is that NTD specifically regulates substrate selectivity (PIP3 vs. PIP2); the dephosphorylation of PIP2s (the second step) can be selectively eliminated by removing NTD a.a. 91/92 and the interacting residues R397 from the PD. Overall, this is an important study with a significant element of novelty, that resolves a previously incompletely understood mechanism. The paper is lucidly written and the illustrations are clear and well explained. I have no major critique; here are several questions and suggestions.

Thank you.

1. Fig. 2 C-E: have you tried the Δ PD construct as the negative control (should eliminate all FRET)?

Thank you for this suggestion. We have added the representative traces and F-V curves of the Δ PD construct co-expressed with FPLC and FTAPP to Fig. 2D, E.

2. I cannot comment much on cryo-EM results, but admittedly the resolution is quite low, especially for the TMD; a brief discussion on the limitations would be helpful.

This is an important point. We did not attempt to build an atomic model of the TMDs due to the limited resolution of the cryo-EM reconstructions in this region. Instead, we only asked whether the AlphaFold predicted structure of Ci-VSP was consistent with the low-resolution map generated from monomeric Dr-VSP particles we classified from the cryo-EM data. We found good agreement between the predicted model and experimentally determined map (Fig. 3E,F), supporting the use of the predicted model as a basis for interrogation of interactions between the VSD, PD, and N-terminal domain. In contrast, the resolution of VSDs from the homodimeric particles we classified was sufficiently high (3.0 Å) to model. These VSDs adopt a resting or inactive conformation similar to previous X-ray structures of isolated VSDs and consistent with the absence of an electric field in the sample. The poorly resolved TMDs in both monomeric and dimeric states are likely a consequence of heterogeneity in their position between particles that results in poor alignment. Attempts to improve TMD alignments with further classification, masking, or signal subtraction were unsuccessful, suggesting alternative approaches to promote discrete conformational states will be required to experimentally determine high resolution structures of TMDs.

3. Just a suggestion for another functional test of $\Delta NT(2-105)$ or the double $D91A/D92A$ mutant: if PIP2 dephosphorylation is absent, the level of PIP2s will at least temporarily rise following the activation of these VSP mutants. Is it possible to use a functional reporter, such a highly PIP2-sensitive channel, to strengthen this conclusion? A temporary enhancement of channel's activity may be expected. Kir2.1Q (R228Q) has an intermediate-small basal current compared to WT Kir1.2 (Zhang et al. 1999) and may be bidirectionally sensitive to PIP2 levels. Alternatively, a partly G $\beta\gamma$ -activated GIRK channel may be responding to an increase in PIP2 in the membrane.

We have compared WT Ci-VSP + Kir2.1Q and $\Delta 2-105$ Ci-VSP + Kir2.1Q to Kir2.1Q alone. As in Fig. S2, we measured PI(4,5)P₂ dephosphorylation using a series of short ramps from a holding potential of -100 mV to follow recovery of PI(4,5)P₂ (and therefore Kir2.1Q current) after a step of depolarization to +60 mV for 100 s that activates Ci-VSP and depletes PI(4,5)P₂. As shown below in Fig. R1, the recovery of current in $\Delta 2-105$ Ci-VSP + Kir2.1Q (seen in the traces) was very small compared to WT Ci-VSP + Kir2.1Q, consistent with almost complete loss of PI(4,5)P₂ dephosphorylation. Moreover, the maximum Kir2.1Q current in $\Delta 2-105$ Ci-VSP + Kir2.1Q was very similar to that of Kir2.1Q alone, suggesting that the high resting level of PI(4,5)P₂ in oocytes is near saturating for the Kir2.1Q mutant.

Figure R1. Readout of $\Delta 2-105$ Ci-VSP effect on PI(4,5)P₂ levels using Kir2.1Q. Experiment performed in two independent rounds with >10 oocytes in each condition in each round. Representative traces shown on the same scale.

4. In the introduction, a clear distinction is made between the gated access model and the PD displacement mode of VSD activation. In the Discussion, it might be helpful to better clarify to the reader whether the findings of this work support/extend one of these models, or both (a synthesis?).

We have modified the end of the Discussion to address this point, as follows:

“The severe impact of the linker/gating loop mutations on VSP function is consistent with the gating access model, although further studies are needed to confirm that gating loop conformational changes catalytic site access. The persistence of some voltage-dependent structural rearrangements and enzyme activity in the D400 mutant may reflect other paths of VSD-PD coupling such as proposed in the PD displacement model.”

Minor points

In Fig. 3 (the AlphaFold2 model) it would be nice to have R397 and the opposing D91/D92 labeled. The side chain of R397 is now shown and labeled.

In Figures 1, 2, and others: in Y axis, Δ shows as \square ; same in Fig. 2 for Δ PD. It looks normal in both Word and pdf, but we will make sure there is no conversion error.

p. 8, sentence starting with “While diffuse density...” has a different font color (dark gray). The font color of that sentence is now the same as the rest.

Methods, p. 14: “the vitelline membrane was removed by collagenase digestion” is incorrect. Collagenase treatment removes outer epithelial and follicular cell layers, leaving the vitelline membrane intact.

Thank you for pointing this out. It has now been changed as follows:

“After *Xenopus laevis* oocytes were harvested, the outer follicular cell layer was removed by collagenase digestion.”

Reviewer #2 (Remarks to the Author):

Dear Editor,

This extensive study by Yawei Yu and Collaborators investigates the mechanism of the operation of voltage-sensing phosphatases (VSP). These enzymes, which possess a transmembrane voltage-sensing domain (VSD) coupled to a cytosolic lipid phosphatase domain (PD), catalyze the dephosphorylation of membrane phosphoinositol phosphates (PIPs) in response to cell depolarization.

*The work addresses the fundamental question of how conformational rearrangements of the voltage-sensing domain propagate to the catalytic site of *Ciona intestinalis* VSP (CiVSP), and attempts to identify the cytoplasmic structures involved in the voltage dependent process.*

The experimental approach is powerful and combines Voltage Clamp Fluorometry (VCF, a technique pioneered by this group) of Ci-VSP, CryoEM of Danio rerio VSP (Dr-VSP) into lipid nanodiscs, FRET-based assessment of enzymatic function, and structural modeling.

By tracking the wave of conformational changes downstream the voltage sensing domain using VCF, the study revealed significant structural rearrangements in three regions: the distal VSD-PD linker, the PD gating loop and the N-terminal domain of the VSD. Interesting CryoEM studies complement the fluorometry information revealing a dimeric structure of the cytoplasmic domain at 3-angstrom resolution and a full-length Dr-VSP at lower resolution.

Abundant experimental and computational work supports a refined model of Voltage-catalysis coupling in VSP. A two-step voltage sensor activation triggers: 1) a reorientation of the VSD-PD linker and the interacting gating loop to open the catalytic for PIP3 and 2) further reorientation of the linker-gating loop and N-terminal domain of the VSD to switch preference for PIP2 at the catalytic site.

Overall, this study advances the field by providing a comprehensive view of the molecular events responsible for the voltage dependence of VSP catalytic activity.

Thank you.

There are some points that need to be addressed or require further clarifications.

Main points:

1) How do the Authors interpret the difference in V-dependence reported by ANAP at positions shown in Fig. S1? Line 139: “Compared to labeling sites 214 in S4 and 243 at the beginning of the linker, the voltage dependence of dye fluorescence (F-V) was shifted positively in the distal linker, gating loop and catalytic site (Fig. S1), suggesting that outward movement of S4 drives subsequent rearrangements that turn on enzyme activity.”

Why does a “positive” shift suggest that S4 drives subsequent rearrangements that turn on enzyme activity?

Could the voltage shift be a manifestation of a cooperative step taking place at the PD? I am asking because CryoEM data have shown that full-length VSP forms dimers.

This is an interesting point and suggests a direction for future study. Our cryo-EM data shows that VSP can exist as a monomer or dimer, but it remains unclear what promotes dimer formation and what the functional consequence is of dimerization. Since our experiments do not address the possible contribution of cooperativity, we are unable to arrive at a conclusion on this. For this reason, we do not take up the possible issue of cooperativity in the Discussion. In this part of the Results, which precedes the presentation of the cryo-EM results, it is awkward to jump ahead and speculate on the potential meaning of a dimer. We therefore confine ourselves here to the tentative suggestion that the voltage dependence shifts indicate sequential rearrangements from S4 to the PD.

2) The collection of ANAP fluorescence at two different wavelengths is a powerful experimental tool: can ANAP spectral changes (as captured in Fig. 1 J) be interpreted more mechanistically also in

view of the structural data from this and previous work? The value of the extensive experimental characterization is sometime lost, such as in the sentence “The reductions in ΔF amplitude and the shifts of ANAP spectrum suggest that the mutations changed either the local environment of the labeling sites or the nature of the rearrangement.”

We have now incorporated the following in the paper:

“To gain insight into how mutations affect the structure at each ANAP labeling site in the resting state, we compared the fluorescence ratio from the two emission channels ($R = F_{460-500} / F_{460-500} + F_{420-460}$) at the holding voltage -100 mV (R_{baseline}). We did not observe any effect of the mutations on R_{baseline} , but low signal to noise ratio due to fluorescence background of the oocytes and/or free ANAP may have obscured a difference. We therefore do not draw a conclusion about this point. However, the analysis of the change of fluorescence ratio between R_{baseline} (- 100 mV) and R_{test} (+ 200 mV) normalized to R_{baseline} ($R_{\text{test}} - R_{\text{baseline}} / R_{\text{baseline}}$, or $\Delta R / R$) eliminates the influence of the background and thus describes the structural changes that each ANAP site undergoes upon activation. The amplitude of $\Delta R / R$ indicates the extent of the spectral shift and the sign of the $\Delta R / R$ shows the direction of the shift. An increase of $\Delta R / R$, such as what we observed with ANAP introduced at position 401 (gating loop residue next to linker-gating loop coupling residue D400) in WT Ci-VSP, indicates a spectral shift to longer wavelengths and exposure to more polar solvent^{21,22}. This is consistent with the overlay of previous X-ray crystal structures of the isolated Ci-VSP PD (**Video-1**), which shows that the side chain of F401 moves to a more solvent exposed environment in the transition from the “closed” state (3V0F, the catalytic site “closed” / E411 blocked conformation) to the “activated” state (3V0H, the catalytic site bound confirmation)¹⁵. The amplitude of the $\Delta R / R$ (i.e. extent of the voltage-dependent spectral shift) was strongly affected by the D400R mutation in the gating loop and the catalytic sites and much less so elsewhere in the protein. Mutations of K252 had even larger effects on the spectral shift (**Fig. 1J**). This is consistent with the severe reduction of coupling to the protein motions that turn on the enzymatic activity in D400R, and the even stronger impact in mutations of K252.”

Lee, H.S., Guo, J., Lemke, E.A., Dimla, R.D. & Schultz, P.G. Genetic incorporation of a small, environmentally sensitive, fluorescent probe into proteins in *Saccharomyces cerevisiae*. *J Am Chem Soc* **131**, 12921-3 (2009).

Chatterjee, A., Guo, J., Lee, H.S. & Schultz, P.G. A genetically encoded fluorescent probe in mammalian cells. *J Am Chem Soc* **135**, 12540-3 (2013).

Liu, L. et al. A glutamate switch controls voltage-sensitive phosphatase function. *Nat Struct Mol Biol* **19**, 633-41 (2012).

3) The co-expression experiments of Ci-VSP and Kir2.1 are good controls to probe the phosphatase activity of ANAP-substituted Ci-VSP constructs (Fig S2). They show that D261- and G409-ANAP enzymatic activity is practically abolished. I am wondering how the results from VCF experiments using these two construct should be interpreted. Please explain.

As pointed out by the reviewer, D261-ANAP and G409-ANAP both have large voltage-driven changes in F (Fig. 1, Fig. S1) and yet, unlike the other ANAP labeling sites, show little depolarization-triggered activity (Fig. S2). In other words, while voltage-activated conformational changes still take place in D261-ANAP and G409-ANAP, mutation to ANAP at these positions disrupts enzymatic function. This is not surprising for the position at gating loop residue G409, which is so close to the E411 gate. As for D261, it is in the same β turn as, and in close proximity to, the linker residues that are important for coupling (K252, R253 and R254), so having ANAP at 261 may

affect linker-gating loop coupling. Because of these perturbations to function, we based our interpretation on other sites where function is preserved despite the introduction of ANAP.

4) Line 133: “ANAP Δ Fs observed from residue 401, near the beginning of the gating loop (Fig. 1E), and a C2 domain residue 555 (Fig. 1H) the homolog of which was suggested to associate with the membrane in PTEN²³, were consistent with a previous study.” Please explain how these VCF data are “consistent with a previous study.”

The referenced paper (Sakata et al., 2016) studied ANAP at these same two sites, 401 and 555, and observed similar Δ Fs in the wildtype background. To clarify this, we have edited the sentence as underlined:

“ANAP Δ Fs observed from residue 401, near the beginning of the gating loop (Fig. 1E), and a C2 domain residue 555 (Fig. 1H), the homolog of which was suggested to associate with the membrane in PTEN²³, were similar as observed previously²⁰.”

5) I did not find methods for statistical analysis (error bars definition, number of replicates, etc.)

Thank you for pointing out this lapse. We have now added this to the Method section under *Data analysis* and sample sizes are reported in individual figure legends.

Other points:

1) Line 223: “Besides the high sequence identity with Ci-VSP (48%) (Fig. S6), *Danio rerio* VSP (Dr-VSP) has an AlphaFold model superposed closely with that of Ci-VSP (RMSD: 0.889 Å) (Fig. 3C), but Dr-VSP with the active site cysteine mutated to serine (C302S) showed better protein expression and stability.” This is an unclear sentence: consider revising it.

We have modified this as follows:

“Compared to Ci-VSP, *Danio rerio* VSP (Dr-VSP) with the active site cysteine mutated to serine (C302S) showed better protein expression and stability. Dr-VSP also has high sequence identity (48%) with Ci-VSP (Fig. S6) and an AlphaFold model superposed closely with that of Ci-VSP (RMSD: 0.889 Å) (Fig. 3C). We therefore incorporated Dr-VSP (C302S) into nanodiscs for the study with cryo-EM (Fig. S7A-D).”

2) Fig 8: this figure shows a background reticle of dotted lines. They seem unnecessary. The dotted lines have been removed.

3) *Fig S1: is missing the x axis.*

The conversion error has been fixed.

4) *Fig S2: Kir recordings have not scale bar (but they do not seem normalized).*

Thank you for pointing it out. Scale bars have been added to all recording traces.

5) *Line 492: missing “e” in baseline*

The typo has been corrected.

6) *Line 705, Fig 3B: Score for low confidence is orange color, not red as indicated in legend.*

The figure legend has been changed to “orange”.

7) *Because of the large number of labeling positions and mutations tested, some parts of the manuscript are difficult to follow. A simple schematic drawing showing the VSP regions discussed (linkers, R/gating loop, catalytic site), mutations and labeling positions, will help to follow the study from the beginning, before more structural information is revealed (Fig. 3).*

Fig. 1 cartoons show each of the labeling sites.

REVIEWERS' COMMENTS

Reviewer #1 (Remarks to the Author):

I am satisfied with the revised version of the paper and the separate answers that the authors have provided for my consideration only. In my opinion, the paper is acceptable in its present form.

Reviewer #2 (Remarks to the Author):

Dear Authors and Editors,

I have reviewed a revised version of the manuscript entitled "Coupling sensor to enzyme in the voltage sensing phosphatase" by et al. Yawei Yu, Lin Zhang, Baobin Li, Zhu Fu, Stephen G. Brohawn and Ehud Y. Isacoff.

The Authors have responded constructively to all my comments, added the methods for statistical analysis and sample sizes and modified the manuscript accordingly.

I have no further questions.